# *OsPMS1* Mutation Enhances Salt Tolerance by Suppressing ROS Accumulation, Maintaining Na^+^/K^+^ Homeostasis, and Promoting ABA Biosynthesis

**DOI:** 10.3390/genes14081621

**Published:** 2023-08-14

**Authors:** Wang-Qing Li, Wen-Jie Zheng, Yan Peng, Ye Shao, Ci-Tao Liu, Jin Li, Yuan-Yi Hu, Bing-Ran Zhao, Bi-Gang Mao

**Affiliations:** 1Longping Branch, College of Biology, Hunan University, Changsha 410125, China; lwq1473639204@163.com (W.-Q.L.); wenjz2022@163.com (W.-J.Z.); 2National Key Laboratory of Hybrid Rice, Hunan Hybrid Rice Research Center, Changsha 410125, China; py225666@163.com (Y.P.); huyuanyi237@163.com (Y.-Y.H.); 3College of Agricultural, Hunan Agricultural University, Changsha 410128, China; 4College of Tropical Crops, Hainan University, Haikou 570228, China; jnli711189@163.com; 5National Center of Technology Innovation for Saline-Alkali Tolerant Rice, Sanya 572000, China

**Keywords:** *OsPMS1*, antioxidant activity, abscisic acid hormone, transcriptome, salt stress

## Abstract

World-wide, rice (*Oryza sativa* L.) is an important food source, and its production is often adversely affected by salinity. Therefore, to ensure stable rice yields for global food security, it is necessary to understand the salt tolerance mechanism of rice. The present study focused on the expression pattern of the rice mismatch repair gene post-meiotic segregation 1 (*OsPMS1*), studied the physiological properties and performed transcriptome analysis of *ospms1* mutant seedlings in response to salt stress. Under normal conditions, the wild-type and *ospms1* mutant seedlings showed no significant differences in growth and physiological indexes. However, after exposure to salt stress, compared with wild-type seedlings, the *ospms1* mutant seedlings exhibited increased relative water content, relative chlorophyll content, superoxide dismutase (SOD) activity, K^+^ and abscisic acid (ABA) content, and decreased malondialdehyde (MDA) content, Na^+^ content, and Na^+^/K^+^ ratio, as well as decreased superoxide anion (O_2_^−^) and hydrogen peroxide (H_2_O_2_) accumulation. Gene ontology (GO) analysis of the differentially expressed genes (DEGs) of *ospms1* mutant seedlings treated with 0 mM and 150 mM NaCl showed significant enrichment in biological and cytological processes, such as peroxidase activity and ribosomes. The Kyoto Encyclopedia of Genes and Genomes (KEGG) metabolic pathway analysis showed that the DEGs specifically enriched ascorbate and aldarate metabolism, flavone and flavonol biosynthesis, and glutathione metabolism pathways. Further quantitative real-time reverse transcription-PCR (qRT-PCR) analysis revealed significant changes in the transcription levels of genes related to abscisic acid signaling (*OsbZIP23*, *OsSAPK6*, *OsNCED4*, *OsbZIP66*), reactive oxygen scavenging (*OsTZF1*, *OsDHAR1*, *SIT1*), ion transport (*OsHAK5*), and osmoregulation (*OsLEA3-2*). Thus, the study’s findings suggest that the *ospms1* mutant tolerates salt stress at the seedling stage by inhibiting the accumulation of reactive oxygen species, maintaining Na^+^ and K^+^ homeostasis, and promoting ABA biosynthesis.

## 1. Introduction

Salinity is one of the most detrimental abiotic stresses affecting crop growth, development, and yield [1]. A high salinity condition affects roughly around 20% of cultivated land worldwide [2]. Various other factors, such as improper irrigation, population growth, industrial pollution, and extreme weather, exacerbate the adverse effects of salt stress on agricultural production [3,4,5]. Rice, one of the major food crops in the world, is moderately sensitive to salt stress during the seedling and reproductive stages [6]. Salt stress can affect the growth of rice and ultimately lead to a reduction in grain yield. Therefore, understanding the response mechanism of rice seedlings to salt stress can provide theoretical support for breeding salt-tolerant varieties, which in turn can help increase the utilization area of saline lands and ensure food security.

In plants under salt stress, the ionic imbalance leads to cell osmotic stress [7]. Osmotic stress leads to rapid closure of stomata, inhibiting carbon dioxide (CO_2_) uptake and weakening photosynthesis [8]. In addition, the accumulation of sodium (Na^+^) and chloride (Cl^−^) ions inside the cells interferes with intracellular uptake and translocation of essential mineral elements, including nitrogen (N), phosphorus (P), potassium (K^+^), calcium (Ca^2+^), and zinc (Zn^2+^) [9]. Since Na^+^ and K^+^ have similar physicochemical properties, Na^+^ at high concentrations competes with the K^+^ binding sites, thereby reducing K-driven enzyme activities and metabolic pathways, such as the Calvin cycle and glycolysis [10,11]. These changes disrupt the intracellular metabolic balance, influencing photosynthetic and respiratory rates and leading to oxidative stress and reactive oxygen species (ROS) accumulation [12]. Finally, these ROS with solid oxidative capacity damage cellular structures and biomolecules (proteins, DNA, and lipids), consequently, inhibiting plant growth [13,14].

Nevertheless, during evolution, plants have developed mechanisms, such as osmotic adjustment, ion homeostasis regulation, reactive oxygen scavenging, and hormone signal transduction, to cope with salt damage. Plants maintain low cell osmotic potential under salt stress by synthesizing and accumulating osmoregulatory substances, such as proline, hydroxyproline, betaine, sugar, and polyamines [11]. In rice under salt stress, high-affinity potassium (K^+^) transporter (HKT1s) and Na^+^/H^+^ exchange protein (NHXs) maintain intracellular Na^+^ homeostasis and reduce ion toxicity [15]. Plants also remove excessive ROS by mobilizing the enzymatic and non-enzymatic components of the antioxidant defense system to prevent membrane peroxidation and maintain membrane stability under adverse conditions [16]. Studies have shown that plants with a strong ability to remove ROS resist salt and drought stresses [17]. Many hormones, such as abscisic acid (ABA), ethylene (ETH), auxin (IAA), gibberellic acid (GA), and jasmonic acid (JA), are involved in regulating the salt stress response and adaptation [18]. Typically, under salt stress, endogenous ABA levels rapidly increase and activate sucrose non-fermentable 1-associated protein kinase 2 protein kinases (SnRK2s), which, in turn, phosphorylate various ABA response element (ABRE)-binding proteins/ABRE-binding factors (AREB/ABF) that regulate ROS clearance, ion homeostasis, and stomatal closure [18,19,20,21].

DNA mismatch repair (MMR) is an important pathway for DNA damage repair, which maintains the genomic integrity of different organisms, and higher plants have MMR systems comprising mismatch repair homologous proteins. Currently, more studies on plant MMR proteins have been reported in *Arabidopsis*. For example, Cao et al. showed that in *Arabidopsis MSH2* and *MSH6* may act as a direct sensor of cadmium-mediated DNA damage [22]. MMR protein msh7 seedlings in *Arabidopsis* showed a reduced salt inhibitory effect [23]. However, the MMR protein in plants under salt stress are rarely reported. A nonsense mutation of the human post-meiotic segregation 2 (*hPMS2*) gene at codon 134 (the *hPMS2-134* mutation) was first identified in hereditary non-polyposis colorectal cancer (HNPCC) patients, and this mutant could cause a dominant negative effect, resulting in MMR deficiency [24]. Chao et al. found that expression of the *hPMS2-134* into *Arabidopsis* inhibited the expression of its endogenous MMR gene and a salt-tolerant phenotype was observed in transformed progeny [25]. Consistent with this observation, in the rice, the overexpression of the truncated version of the protein OsPMS1-136 (1-136 amino acids of OsPMS1) resulted in salt-tolerant and drought-resistant mutant phenotypes in progeny transgenic plants [26]. In this study, we constructed the knockout lines of *OsPMS1*, a critical gene in the plant DNA MMR system. The phenotype of *ospms1* mutant rice seedlings was identified with 150 mM NaCl, and the salt tolerance mechanism of *OSPMS1* was initially analyzed from physiological characteristics and transcriptome sequencing, which provided a theoretical basis for breeding salt-tolerant rice varieties.

## 2. Materials and Methods

### 2.1. Plant Materials and Growing Conditions

The CRISPR/Cas9 genome editing technique was used to generate *ospms1* mutant rice in the Huazhan background. The short guide RNAs (sgRNAs) were targeted to exons 1 and 2 of *OsPMS1* according to the previously described method [27], using the target-linked primers Cas9-OsPMS1-F and Cas9-OsPMS1-R and the target-linked primers Cas9-F and Cas9-R to construct the knockout vectors pYLCRISSPR/Cas9-MT(I)-OsPMS1, respectively. Appendix A contains the primer pairs used in the study.

Seeds from wild-type Huazhan and the T3 generation *ospms1* mutants (*ospms1-1* and *ospms1-2*) were used in this study. First, these seeds were washed 3 times with sterile water, sterilized with 75% (*w*/*v*) ethanol for 2 min, and shaken with 1% sodium hypochlorite solution for 20 min. Finally, they were rinsed 5 times with sterile distilled water before germination at 37 °C and 60% relative humidity for 1 day. Then, the germinated seeds were placed into bottomless 96-well plates filled with Yoshida’s solution and grown for 12 days, and 12-day-old seedlings were transferred to a Yoshida’s solution containing 150 mM NaCl for 10 days. Subsequently, they were transferred to Yoshida’s solution for 7 days of recovery growth, and finally, the survival rate was calculated based on the percentage of viable seedlings. The solution should change every 3 days during seedling culture. All seedlings were cultured in an artificial climate chamber at 30 °C with 70% relative humidity with a 12 h/12 h (light/dark) photoperiod (Changsha, Hunan).

### 2.2. Gene Structure and Protein Sequence

The rice database (https://www.ricedata.cn/gene/ accessed on 8 August 2021) provided information regarding *OsPMS1* genomic sequence, CDS sequence, and protein sequence. We analyzed the exon-intron structures of *OsPMS1* using Gene Structure Display Server 2.0 (GSDS, https://gsds.cbi.pku.edu.cn/ accessed on 8 August 2021). The NCBI’s BlastP website (https://blast.ncbi.nlm.nih.gov/Blast.cgi accessed on 15 August 2021) was used to download the amino acid sequences of ZmPMS1, HvPMS1, SbPMS1, AtPMS1, HsPMS1, MuPMS1, SpPMS1 homologues, and we used DNAMAN software to align amino acid sequences. MEGA6.0 software was used to construct phylogenetic trees with the neighbor-joining (NJ) method and 1000 bootstrap replications. In order to analyze the conserved domains of proteins, the website http://www.ncbi.nlm.nih.gov/Structure/cdd/wrpsb.cgi (accessed on 15 August 2021) was used.

### 2.3. Subcellular Localization of OsPMS1

To determine the intracellular localization of OsPMS1 protein, we constructed the OsPMS1-GFP expression vector by amplifying the OsPMS1 coding region by means of PCR and inserting it into the pYBA1132-GFP vector containing the CaMV 35S promoter. Then, the obtained OsPMS1-GFP plasmid and positive control of the green fluorescent protein (GFP) were co-expressed with nuclear markers (AtWRKY25-mCherry) in *Nicotiana benthamiana* leaves or rice protoplasts, respectively [28]. A confocal scanning microscope (LSM 880, Zeiss, Jena, Germany) was used to observe the GFP signal (green) and an mcherry signal (red). GFP was excited with a laser line of 488 nm and detected at a wavelength of 510 nm. The mCherry was excited at 587 nm, and emissions were detected at 610 nm.

### 2.4. NBT and DAB Staining

The methods described by Wang et al. [29] utilized nitro blue tetrazolium chloride (NBT) and diaminobenzidine (DAB) staining to detect cellular accumulation of superoxide anion (O_2_^−^) radicals and hydrogen peroxide (H_2_O_2_) in rice leaves, with slight modifications. The collected rice seedling leaves were under 0 and 3 days of salt stress in 150 mM NaCl. For NBT staining, plant leaves were first placed in a 5 mL test tube with NBT staining solution and vacuumed for 30 min until the solution was completely immersed in the leaves and then stained in the dark for 12 h. For DAB staining, plant leaves were first put into a 5 mL test tube containing DAB staining solution, vacuumed for 1 h, and then stained for 24 h in a 37 °C incubator. Subsequently, the stained samples were boiled in 95% (*w*/*v*) ethanol to remove chlorophyll and washed twice with deionized water to observe and photograph.

### 2.5. Measurement of Physiological Indexes

In order to analyze the changes in physiological indexes of the mutants in response to salt stress, on the 0 and 3 days of salt stress treatment, the seedling leaves were taken to determine physiological indexes.

#### 2.5.1. Measurement of Superoxide Dismutase

For the determination of superoxide dismutase (SOD), first, added rice leaves (0.5 g) were added to 1 mL phosphate buffer solution (pH 7.2) with a mortar for ice bath homogenization, and then the homogenate was centrifuged at 8000× *g* at 4 °C for 10 min. The obtained supernatant was crude enzyme extract. SOD activity was determined according to the test kit’s instructions (ZCIBIO Technology Co., Ltd., Shanghai, China; No: ZC-SO350).

#### 2.5.2. Measurement of Malondialdehyde

The rice leaves (0.5 g) were ground to powder form with liquid nitrogen, and, then 5 mL of 10% trichloroacetic acid (TCA) was added and mixed well, and the supernatant was collected after centrifugation at 5000 rpm for 10 min. After mixing 2 mL of the supernatant well with 2 mL of 0.6% (*w*/*v*) thiobarbituric acid (TBA), the mixture was boiled for 15 min and centrifuged at 12,000 rpm for 10 min at 4 °C [17]. Finally, the supernatant was collected, and the microplate readers with full wavelength (Spectra Max190, Molecular Devices, San Jose, CA, USA) were used to measure absorbance at wavelengths 450, 532, and 600 nm, respectively.

#### 2.5.3. Measurement of Relative Chlorophyll Content

A SPAD-502 chlorophyll meter (Konica Minolta, Tokyo, Japan) was used to measure relative chlorophyll content. Firstly, SPAD-502 readings for each leaf were averaged from three places in the upper, middle, and lower parts. Finally, each rice accession SPAD reading was calculated, based on the average SPAD value of the leaves of three randomly selected plants [30].

#### 2.5.4. Measurement of Relative Water Content

Six rice seedling leaves were taken as a group, weighed and their fresh weight (Fw) was recorded, the leaf samples were then soaked in distilled water for 8 h to ensure that the leaves fully absorbed the water and reached turgid weight (Tw), and weigh and record the turgid leaves weighed and the weight recorded after fully absorbing the surface moisture with absorbent paper. Finally, all samples were placed in an oven at 80 °C, baked to constant weight, and the dry weight (Dw) was recorded [31]. We calculated the relative water content (RWC) using the formula: RWC (%) = (Fw − Dw)/(Tw − Dw) ×100.

#### 2.5.5. Measurement of Soluble Protein Content

First, 0.1 g of rice leaves was weighed and ground into homogenate in 1 mL precooled phosphate buffer with a pH of 7.2, and, then, transferred to a centrifuge tube and centrifuged at 1000 rpm at 4 °C for 20 min. The supernatant obtained was the crude protein extract. The content of soluble protein was measured using the Coomassie Brilliant Blue method, described by Bradford [32].

#### 2.5.6. Determination of Na^+^ and K^+^ Content

The content of Na^+^ and K^+^ was measured according to the method described by predecessors [33], with minor modifications. The leaves were washed 3 times with deionized water and baked at 80 °C for 3 days to constant weight. All samples were digested with nitric acid for 12 h and heated at 200 °C for about 8 h. Finally, we diluted all samples with deionized water and performed inductively coupled plasma mass spectrometry (ICP-MS) for Na^+^ and K^+^ in the solution.

#### 2.5.7. Abscisic Acid Determination

ABA content was measured according to the plant hormone abscisic acid enzyme-linked immunosorbent assay kit (ZCIBIO Technology Co., Ltd., Shanghai, China; No: ZC-53364), as previously described [34]. Approximately 0.1 g of rice seedling leaves were grounded into a fine powder with liquid nitrogen. Then, the powder was transferred to a centrifuge tube containing 1 mL 80% methanol and, after incubation overnight at −20 °C, the samples were centrifuged at 4 °C at 8000 rpm for 15 min. The supernatant was passed through at C-18 solid phase extraction column, and the sample was vacuum dried after passing through the column. Finally, 1 mL of methanol Tris-HCl buffer (pH 7.4) was added and was centrifuged at 8000 rpm for 15 min at 4 °C, and the microplate readers with full wavelength (Spectra Max190, Molecular Devices, San Jose, CA, USA) were used to measure the absorbance of the supernatant at 450 nm wavelength, and the ABA content was calculated according to the standard curve and formula provided in the kit.

### 2.6. Transcriptome Sequencing Analysis

To preliminarily explore the molecular basis of salt tolerance in *OsPMS1*, transcriptomic sequencing was performed on the leaves of wild-type and *ospms1* mutant seedlings exposed to 0 and 6 h of salt stress.

#### 2.6.1. RNA Extraction, Library Construction, and Sequencing

Total RNA was extracted from seedling leaves using an RNA pure Plant kit (Magen, China). The quality and concentration of total RNA were measured by means of agarose gel electrophoresis and NanoDrop 2000 (Thermo Scientific, Waltham, MA, USA). The integrity of total RNA extraction was further analyzed using Agilent Bioanalyzer 2100 (Agilent, CA, USA). Each qualified RNA sample was taken for RNA-seq and qRT-PCR analysis. The strand-specific library was constructed with extracted RNA, and double-ended sequencing was performed using an IluminaHiseg-PE150 high-throughput sequencer. Beijing Novogene Bioinformation Technology Co., Ltd. (Beijing, China) completed the library construction and sequencing.

#### 2.6.2. Enrichment Analysis of Differentially Expressed Genes (DEGs) and Their GO and KEGG Pathways

The SOAP nuke software (BGI, Shenzhen, China) was used to remove reads with un-known bases (>5%) and low-quality reads (quality values < 10 and more than 20% of the total number of bases read) in order to obtain a clean read [35]. Then the clean reads to the reference genome (*Oryza sativa* Group 4.0, https://www.ncbi.nlm.nih.gov/nucore/255672756?report=fasta accessed on 12 December 2021), and the gene expression levels were quantified by quantifying cDNA fragments per kilo bases per million mapped reads (FPKM). DESeq2 software was used to identify DEGs. DEGs were defined as the *p*-value < 0.05 and fold change > 2. The gene ontology (GO) and Kyoto Encyclopedia of Genes and Genomes (KEGG) pathway enrichment (http://geneontology.org/ accessed on 26 December 2021) of DEGs were analyzed using gene ontology resources, and the Tbtools software was used to plot the FPKM expression values of the differentially expressed gene list in a heatmap [36].

#### 2.6.3. Quantitative Real-Time Reverse Transcription-PCR (qRT-PCR)

Nine differentially expressed genes from the transcriptome differential genes that were significantly qRT-PCR were selected to validate the RNA-seq results. RNA was reverse-transcribed to complementary DNA using the Super Script II kit (TaKaRa, Shuzo, Japan). The qRT-PCR was performed using the Roche LightCycler 480 II instrument in combination with ChamQ Universal SYBR qPCR Master Mix (Vazyme Biotech Co., Ltd., Nanjing, China). We calculated the relative expression levels of genes using the 2^−∆∆CT^ method [37]. Appendix A contains primers for qRT-PCR.

### 2.7. Statistical Analysis of Experimental Data

All data in this paper were expressed as mean ± standard deviation (SD). The error bars indicate the SD. All experiments had at least three independent biological replicates. Student’s *t* test was used to determine whether there were significant differences between the two groups.

### 2.8. Data Availability Statement

The RNA-Seq data from this study can be found in the online repository. The Biological Project Accession Number in NCBI is PRJNA974431.

## 3. Results

### 3.1. Expression Analysis and Subcellular Localization of OsPMS1

The full length of the *OsPMS1* genomic sequence is 5953 bp and it has a CDS of 2772 bp, with 11 introns and 12 exons. The *OsPMS1* gene encodes a 923 amino acid (aa)–long protein with a predicted molecular weight of 101 kD and three conserved functional domains, as follows: HATPase_c at the N-terminus (24–106 aa) and the DNA_mis_ repair (221–341 aa) and the MutL_c domains (719–876 aa) at the C-terminus. Phylogenetic analysis using the amino acid sequences of OsPMS1 and three *Gramineae* plants (ZmPMS1, HvPMS1, SbPMS1), *Arabidopsis thaliana* (AtPMS1), *Homo sapiens* (HsPMS1), *Mus musculus* (MuPMS1), and *Schizosaccharomyces pombe* (SpPMS1), and OsPMS1 was found to have 69.39% identity with barley HvPMS1 (Appendix A). We further investigated the tissue-specific and salt stress-induced expression profile of *OsPMS1* by means of qRT-PCR. We found that *OsPMS1* was expressed in six tissues of rice and was highly expressed in spikes and leaves. (Figure 1A). The qRT-PCR analysis showed that the expression of *OsPMS1* can be induced by salt stress (Figure 1B). The relative expression level of *OsPMS1* increased dramatically at 6 h and peaked at 12 h. Then, confocal microscopy detected that, in *Nicotiana benthamiana* leaves and rice protoplasts, the GFP signal coexisted with the AtWRKY25-mCherry signal, indicating that OsPMS1 is a nuclear protein (Figure 1C,D).

### 3.2. OsPMS1 Knockout Mutants Exhibited Salt Tolerance

The knockout lines of *OsPMS1*, one with a 1 bp insertion and a 31 bp deletion (*ospms1-1*) and another with a 1 bp insertion and a 1 bp deletion (*ospms1-2*), both resulted in frameshift mutations in the OsPMS1 protein (Figure 2A). The T3 generation *ospms1* mutant (*ospms1-1* and *ospms1-2*) seedlings showed no significant difference in morphology compared with the wild-type seedlings at day 12. Subsequently, we treated these 12-day-old rice seedlings with 150 mM NaCl to observe their growth. On the 5th day of treatment, the wild-type seedlings exhibited obvious wilt symptoms on almost all leaves, while the *ospms1* mutant seedlings had only a few withered leaves. On the 10th day of treatment, the leaves of the *ospms1* mutant seedlings appeared less shrunk and curled than the wild-type plant; the stems of these mutants were greener and firmer, with higher fresh and dry weights than those of the wild-type seedlings (Appendix A). Collectively, the *ospms1* mutants showed better growth than the wild-type seedlings (Figure 2B). The survival rate of the two *ospms1* mutant seedlings were significantly higher than that of the wild-type seedlings (Figure 2C). Then, we selected the *ospms1-2* mutant, with a more pronounced and stable salt-tolerant phenotype, to assess the relative water content and the relative chlorophyll content. We found that the relative water content and the relative chlorophyll content of the *ospms1-2* mutant seedlings were significantly higher than those of the wild-type seedlings (Figure 2D,E). These results indicated that *ospms1* mutant seedlings had a better salt-tolerant phenotype than the wild-type seedlings at the seedling stage.

### 3.3. OsPMS1 Mutation Enhanced ROS Scavenging under Salt Stress

No significant difference was seen in the soluble protein content, MDA content and SOD activity before salt treatment. After salt treatment, the soluble protein content, MDA content, and SOD activity of the wild-type and mutant seedlings increased. Compared with the wild-type seedlings, the mutants showed significantly reduced MDA content and significantly increased SOD activity; however, no significant difference was observed in the soluble protein content (Figure 3A–C). Subsequent NBT and DAB staining showed no noticeable staining in these leaves before salt treatment, but a more distinct blue and dark brown color in the wild-type leaves than in *ospms1* mutant leaves after 3 days of salt treatment (Figure 3D). These observations indicated a higher ability of the *ospms1* mutants to scavenge ROS, such as superoxide anion (O_2_^−^) and hydrogen peroxide (H_2_O_2_), than the wild-type seedlings. In conclusion, the *ospms1* mutants protected themselves from salt stress by removing the accumulated ROS.

### 3.4. OsPMS1 Mutants Maintained Na^+^ and K^+^ Homeostasis under Salt Stress

Under normal conditions, no significant differences were detected in Na^+^ and K^+^ levels in the leaves of the *ospms1* mutants and wild-type seedlings. After salt treatment, the Na^+^ content in the leaves of the wild-type and *ospms1* mutants increased (Figure 4A), the K^+^ content decreased (Figure 4B), and the Na^+^/K^+^ ratio increased (Figure 4C). Compared with the wild-type seedlings, the K^+^ content of the leaves of the *ospms1* mutants increased significantly under salt stress, and the Na^+^ content and the Na^+^/K^+^ ratio decreased significantly. These observations suggest that *ospms1* mutants tolerate salt stress by adjusting the Na^+^/K^+^ ratio and maintaining the balance between Na^+^ and K^+^ in cells.

### 3.5. OsPMS1 Mutation Promoted ABA Synthesis under Salt Stress

We found no significant difference in ABA content between wild-type seedlings and *ospms1* mutant seedlings under normal conditions. However, after three days of salt stress, the ABA content in all plants increased, and the ABA content in *ospms1* mutants increased significantly compared to wild-type seedlings (Figure 4D). This result suggests that *ospms1* mutant seedlings adapt to salt-stressed environments by promoting ABA biosynthesis.

### 3.6. Transcriptome Analysis of Salt-Treated ospms1 Mutant and Wild-Type Seedlings

The analysis generated 4.5–6.8 million clean reads, with a Q30 value of more than 92% and a GC content of more than 53% (Appendix A), indicating good quality of the RNA sequencing data. Statistical analysis showed that 5764 DEG, were produced in wild-type seedlings (marked as group SWT vs. NWT), of which 2762 genes were upregulated, respectively, and 3002 genes were down-regulated. On the other hand, the *ospms1* mutant seedlings had 5000 DEGs (marked as group SP1 vs. NP1), of which 2334 were upregulated and 2666 genes were downregulated (Figure 5A,B). The Venn plot found 3192 genes shared among the SWT vs. NWT and the SP1 vs. NP1 groups (Figure 5C). These results indicated a change in the genes of *ospms1* mutant seedlings at the transcription level under salt stress, which could alleviate the harm caused by salt stress.

Further, GO functional enrichment analysis of the DEGs from the SWT vs. NWT and SP1 vs. NP1 groups were annotated to “biological processes”, “cellular components”, and “molecular functions” (Figure 6A,B). In the molecular function category, DEGs of both SWT vs. NWT and SP1 vs. NP1 groups were enriched for GO entries of oxidoreductase activity, anion binding, and glycosyltransferase activity, while those of the SP1 vs. NP1 group were also specifically enriched in peroxidase activity and oxidoreductase activity acting on peroxide as acceptor. In the biological process category, both SWT vs. NWT and SP1 vs. NP1 DEGs were enriched in response to stimulus, cell wall organization, and the oxidation reduction process, and the SP1 vs. NP1 DEGs were also specifically enriched to regulate cellular and biological processes. In the cellular components category, both the SWT vs. NWT and SP1 vs. NP1 DEGs were enriched to membrane and chloroplast, and the SP1 vs. NP1 DEGs were also specifically enriched to the ribosome. These results suggest that the *ospms1* mutant seedlings enhance salt tolerance by mobilizing oxidative stress, cellular metabolism, bioregulation, and ribosome-related gene expression.

We performed KEGG metabolic pathway enrichment analysis to determine the metabolic pathway responsible for increased salt tolerance in *ospms1* mutants. The DEGs obtained from the SWT vs. NWT group and the SP1 vs. NP1 group significantly enriched metabolic pathways and biosynthesis of secondary metabolites (Figure 7A,B). The DEGs of SP1 vs. NP1 and SWT vs. NWT groups enriched plant hormone signal transduction, MAPK signaling pathway, photosynthesis-antenna proteins, porphyrin, and chlorophyll metabolism pathways. In addition, a few DEGs in the SP1 vs. NP1 group specifically enriched ribosome, DNA replication, ascorbate and aldarate metabolism, flavone and flavonol biosynthesis, glutathione metabolism, and carotenoid anabolic pathways. These results suggest the significance of these pathways in salt stress response and tolerance mechanisms in the *ospms1* mutants.

According to the results of GO enrichment and KEGG pathway enrichment, we screened out nine candidate genes related to the salt stress response, which are shown in Figure 5B. We divided these genes into those associated with plant hormone signaling (*OsbZIP23*, *OsSAPK6*, *OsNCED4*, *OsbZIP66*, *SIT1*), ROS homeostasis (*OsTZF1*, *OsDHAR1*), ion transport (*OsHAK5*), and osmoregulation (*OsLEA3-2*). The heat map based on the RNA-seq data clearly showed the transcription levels of these DEGs (Figure 8A). Finally, to verify the reliability of our RNA-seq data, we measured the transcription levels of these nine DEGs using qRT-PCR. As shown in Figure 8B–J, the expression trends of all these genes were similar to those obtained from the sequencing data, indicating the reliability of the transcriptome sequencing data.

## 4. Discussion

The MMR protein has been shown to respond to cadmium stress in rice, *Arabidopsis*, and *soybean* [22,38]. Still, there are limited reports on the role of rice MMR protein in response to salt stress. Xu et al. eventually obtained salt-tolerant transgenic plants by overexpressing a truncated OsPMS1-136 protein in rice, which caused MMR defects through dominant negative repression [26]. However, the exact mechanism of salt tolerance is not known. In our study, OsPMS1 is a nucleoprotein mainly expressed in spikes and leaves and rapidly upregulated under salt stress. We used the CRISPR/Cas9 system to directly knockdown the rice OsPMS1 protein, and the pure *ospms1* mutant seeds showed a salt-tolerant phenotype when treated with salt stress at seedling stage. We then preliminarily elucidated the response mechanism of this mutant to salt stress at the seedling stage by physiological characterization and transcriptome sequencing.

Maintaining water status regulates cell expansion, elongation, and division during plant growth [31]. Researchers use the relative water content of leaves as one of the efficient methods for evaluating rice cultivars for tolerance to salt stress [39]. Besides, under adversity, the chlorophyll level reflects the plant’s photosynthetic function, and high chlorophyll content improves the salt tolerance of rice [40,41]. Our analysis revealed that the *ospms1* mutant exhibited some significant phenotypes under salt stress, including higher fresh weight, dry weight, survival rate, relative water content, and chlorophyll content, compared to the wild-type seedlings. Our study demonstrated that *OsPMS1* negatively regulates rice salt tolerance.

Soluble protein is a vital osmotic regulator and nutrient of plants. This component can improve the cell water retention capacity, and can protect the life substances and biofilms. Plants carry out osmotic regulation by accumulating soluble proteins to reduce the osmotic potential and alleviate the ionic toxicity of salt stress [42]. Interestingly, the *ospms1* mutant also exhibited specific enrichment of structural genes encoding ribosomes. Therefore, it is speculated that the *ospms1* mutant might maintain normal physiological functions by promoting cell protein synthesis. In addition, the *ospms1* mutant seedlings showed significant *OsLEA3-2* upregulation compared to wild-type seedlings under salt stress. Proteins from the late embryogenesis abundant (LEA) superfamily are expressed in response to various environmental stresses during plant development and play an essential role in improving plant stress tolerance. Studies have shown that an overexpression of *OsLEA3-2* improved drought tolerance and salinity tolerance [43]. These findings collectively suggest that the *ospms1* mutants regulate the expression of *OsLEA3-2* to increase the accumulation of osmotic adjustment substances and, thereby, improve salt tolerance.

Maintaining intracellular ion balance is necessary to ensure metabolic processes and normal development. Under salt stress, plants usually promote Na^+^ absorption but inhibit K^+^ absorption [44,45]. In our study, the *ospms1* mutant had higher K^+^ content and lower Na^+^ content and Na^+^/K^+^ ratio than the wild-type seedlings after salt stress exposure. Increasing the K^+^/Na^+^ ratio can enhance the salt tolerance of plants. Earlier, Yang found that the overexpression of *OsHAK5* increased the K^+^/Na^+^ percentage and improved the salt tolerance of plants. On the contrary, the K^+^/Na^+^ ratio of the *OsHAK5* knockout mutant decreased, increasing the susceptibility to salt stress [46]. Our results are consistent with these earlier findings. After salt stress exposure, *OsHAK5* expression in *ospms1* mutant seedlings was significantly increased, compared with wild-type seedlings. In addition, GO enrichment analysis showed that a number of ion transport related genes were significantly enriched under salt stress. These results suggest that the *ospms1* mutant regulated the expression of ion transport-related genes to maintain intracellular Na^+^/K^+^ balance, thereby improving salt tolerance.

Salt stress-induced osmotic stress and ionic stress significantly increase ROS levels and results in oxidative stress [47]. Plants eliminate excessive ROS produced by salt-induced stress by activating the enzyme system, which increases intracellular antioxidant enzyme activity [48]. Malondialdehyde is an indicator of lipid peroxidation. The stronger the lipid peroxidation, the higher the malondialdehyde content [49]. Our study found that after exposure to salt stress, the *ospms1* mutant had significantly lower malondialdehyde content and a highly significant higher SOD activity than those in wild-type seedlings. Interestingly, NBT and DAB staining also showed less O_2_^-^ and H_2_O_2_ in the cells of the *ospms1* mutant. Our transcriptome analysis showed DEG-specific enrichment to antioxidant metabolic pathways, such as ascorbate and aldarate metabolism, flavone and flavonol biosynthesis, and glutathione metabolism in the *ospms1* mutant. RNA-seq and qRT-PCR detected that the expression levels of *OsTZF1* and *OsDHAR1* in *ospms1* mutant after salt stress were higher than those of wild-type seedlings, and the expression level of *SITI* was significantly lower than that of wild-type seedlings. Studies have shown that the tandem zinc finger protein *OsTZF1* enhances the salt tolerance of rice by regulating ROS homeostasis and other stress-related genes [50]. Meanwhile, *OsDHAR1* overexpression in rice improved the tolerance to salt stress by neutralizing ROS [51]. Ethylene is another abiotic stress signal, and *SIT1* mediates ethylene signal transduction and ROS accumulation by phosphorylating MPK3/6 [52]. These observations suggest that the *ospms1* mutant efficiently removes excess ROS and resist it salt stress.

Abscisic acid is another crucial factor that plays a vital role in plant growth and stress adaptation. The present work found that the content of ABA in the ospms1 mutant increased significantly under salt stress. The enzyme 9-cis-epoxycarotenoid dioxygenase (NCED) was considered to be the key rate-limiting enzyme in the biosynthetic pathway of ABA [53]. Additionally, the DEGs in the *ospms1* mutant under salt stress were specifically enriched in the carotenoid anabolic pathway. In addition, in the *ospms1* mutant, RNA-seq data and qRT-PCR confirmed that ABA hormone-related genes (*OsbZIP23*, *OsSAPK6*, *OsNCED4*, *OsbZIP66*) were significantly upregulated. *OsSAPK6* belongs to the SnRK2 protein family and has been reported to be involved in the ABA signaling pathway [54]. Zong et al. demonstrated that *OsSAPK6* can phosphorylate *OsbZIP23* and induce *OsbZIP23* transcriptional activation in vivo, which in turn can positively regulate *OsNCED4*, a key gene for ABA biosynthesis [55]. In addition, the bZIP transcription factors *OsbZIP66* and *OsbZIP23* were also participate in the ABA signaling pathway, and transgenic rice overexpressing *OsbZIP23* showed significantly increased tolerance to drought and high salt stresses and significantly increased sensitivity to ABA [56]. These results suggest that under salt stress, *ospms1* mutants may adapt to an adverse environment by promoting ABA biosynthesis and activating the expression of genes related to the ABA signaling pathway.

## 5. Conclusions

In summary, we focused on the physiological and molecular basis of *ospms1* mutant seedlings under salt stress. Physiological analyses showed that the *ospms1* mutant enhanced salt tolerance by suppressing ROS accumulation, maintaining Na^+^/K^+^ homeostasis and promoting ABA biosynthesis. The transcriptome analysis indicated that genes related to ROS homeostasis, the ABA signaling pathway, and osmotic and ionic homeostasis may be involved in salt stress tolerance in *ospms1* mutant seedlings. Therefore, we preliminarily proposed a working model of *ospms1* mutant response to salt stress at the seedling stage (Figure 9). In conclusion, the rice mismatch repair protein OsPMS1 negatively regulates salt tolerance in rice. This study initially explored the mechanisms involved in the regulation of salt tolerance in rice, providing an important theoretical basis for molecular breeding for salinity tolerance in rice, but the exact molecular mechanism of *OsPMS1* response to salt stress still needs to be further elucidated.

## Figures and Tables

**Figure 1 genes-14-01621-f001:**
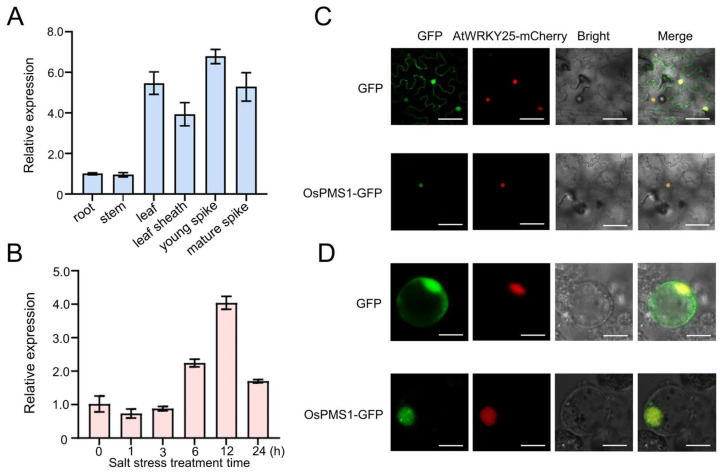
Expression pattern and subcellular localization of *OsPMS1*. (**A**) Quantitative real-time reverse transcription-PCR (qRT-PCR) was performed to analyze the expression levels of *OsPMS1* in different tissues. (**B**) Transcript levels of *OsPMS1* were induced by 150 mM NaCl in 12-day-old seedlings. Subcellular localization of OsPMS1-GFP in *N. benthamiana* leaves (**C**), and rice protoplast (**D**) From left to right, the images are 35S::GFP control vector, AtWRKY25-mCherry: a marker anchored in the cell nucleus, Bright field, and Merged images. Scale bar = 20 µm. Data are shown as mean ± SD (*n* = 3).

**Figure 2 genes-14-01621-f002:**
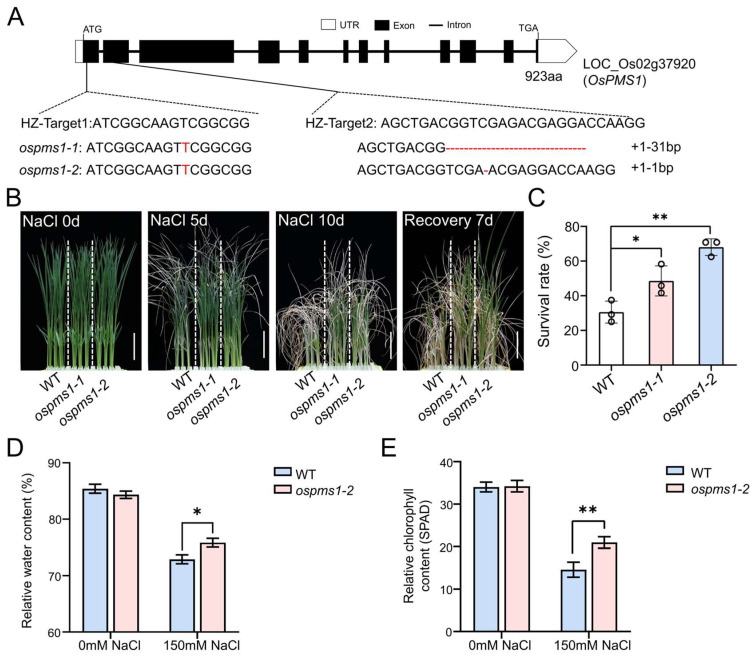
Mutants of *ospms1* were conferred salt tolerance. (**A**) Gene structure, two CRISPR/Cas9 target positions, and mutation types. UTR, exon, and intron are indicated by blank rectangle, black rectangle, and black line, respectively. (**B**) The seedlings of WT, *ospms1-1* mutant and *ospms1-2* mutant grown under 12 h light/12 h dark conditions for 12 days (left panel), transferred to 150 mM NaCl for 5 and 10 days (the two middle panels) and recovered for 7 days (right panel), respectively. Scale bar = 5 cm. (**C**) Statistical analysis of the survival rate after 7 days of recovery; (**D**) relative water content and relative chlorophyll content (**E**) at 0 and 3 days of salt stress. Data are presented as mean ± SD. (*n* from 3 biological replicates and 24 plants were tested in each of biological replicates) Students’ *t* test was used to determine statistical significance (* *p* < 0.05, ** *p* < 0.01).

**Figure 3 genes-14-01621-f003:**
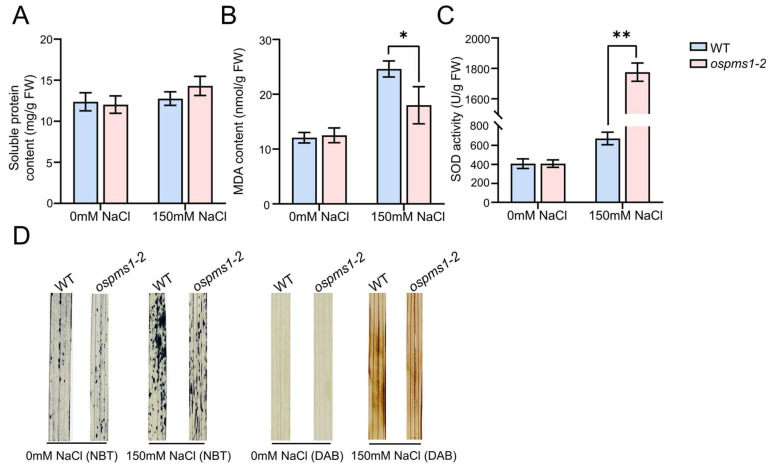
Oxidative stress in leaves of *ospms1-2* mutant and WT seedlings at 0 and 3 days of salt stress. (**A**) soluble protein content. (**B**) MDA content. (**C**) SOD activity; Each column represents the mean ± SD (three replicates). Significant differences, which were determined by Students’ *t* test, are indicated by asterisks (* *p* < 0.05 and ** *p* < 0.01). (**D**) ROS detection in the leaves of WT and *ospms1-2* mutants under normal and salt stress conditions. Leaves stained with NBT and DAB were used to assess O_2_^−^ and H_2_O_2_ accumulation, respectively. Images were taken 0 and 3 days of treatment with 150 mM NaCl. (There were three independent biological replicates, with each biological replicate containing 15 independent plants).

**Figure 4 genes-14-01621-f004:**
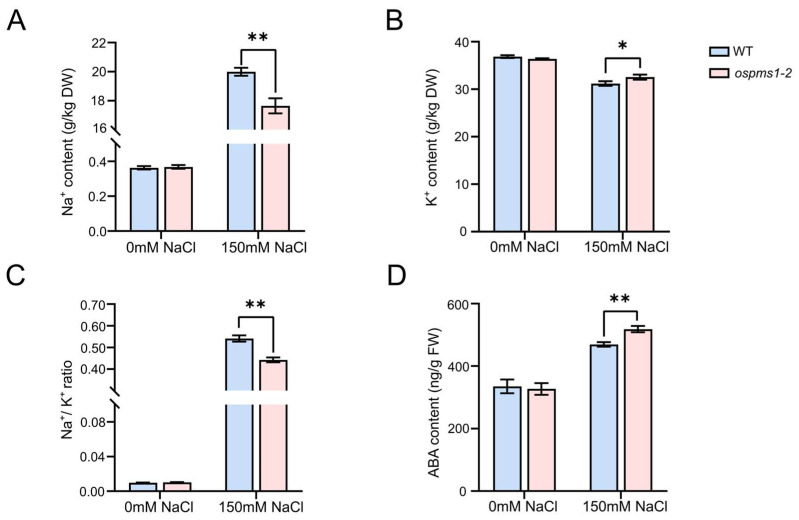
The content of Na^+^, K^+^, and ABA in leaves of *ospms1-2* mutant and WT rice seedlings at 0 and 3 days under salt stress. (**A**) Na^+^ content. (**B**) K^+^ content. (**C**) Na^+^/K^+^ ratio. (**D**) ABA content. Each column represents the mean ± SD (three replicates). Significant differences, which were determined by Students’ *t* test, were indicated by asterisks (* *p* < 0.05 and ** *p* < 0.01).

**Figure 5 genes-14-01621-f005:**
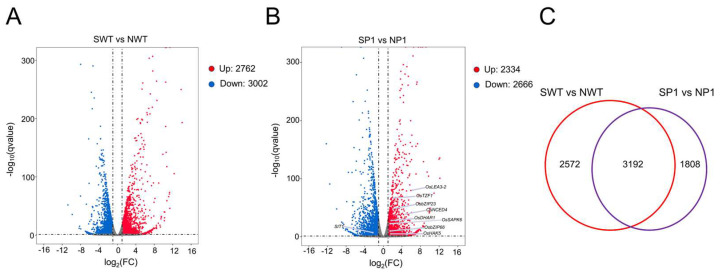
Selection of differentially expressed genes in WT and *ospms1-2* mutant before and after salt stress. (**A**) Volcano plots of differentially expressed genes in WT rice seedlings before and after salt stress. (**B**) Volcano plots of differentially expressed genes in *ospms1-2* mutant rice seedlings before and after salt stress. (**C**) Venn diagram of differentially expressed genes in WT and *ospms1-2* mutant rice seedlings before and after salt stress.

**Figure 6 genes-14-01621-f006:**
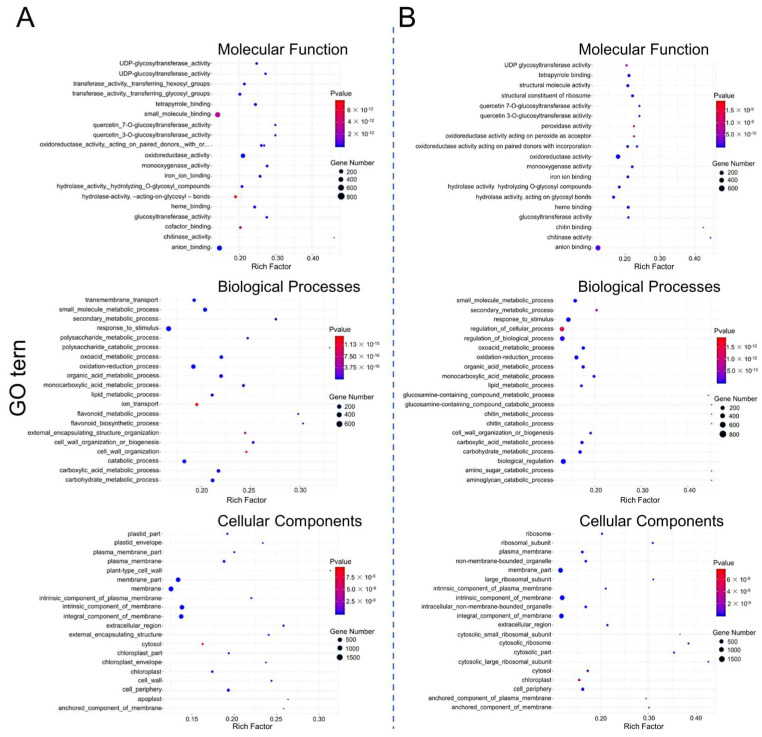
GO enrichment analysis of DEGs of WT (**A**) and *ospms1-2* mutant (**B**) at 0 and 6 h of salt stress. The *Y*-axis of the bubble diagram represents GO enrichment terms in molecular functions, biological processes, and cellular components, and the *X*-axis represents the enrichment fraction. The larger the bubble, the greater the number of DEGs involved. The redder the color of the bow, the more significant the enrichment effect.

**Figure 7 genes-14-01621-f007:**
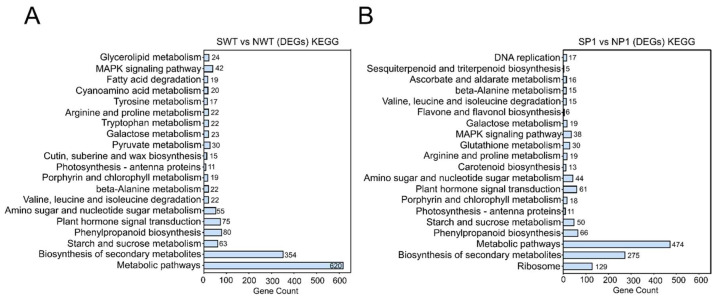
KEGG pathway enrichment analysis of DEGs of WT (**A**) and *ospms1-2* mutant (**B**) at 0 and 6 h of salt stress. The *Y*-axis of the bar indicates the KEGG pathway, and the *X*-axis shows the number of enriched genes.

**Figure 8 genes-14-01621-f008:**
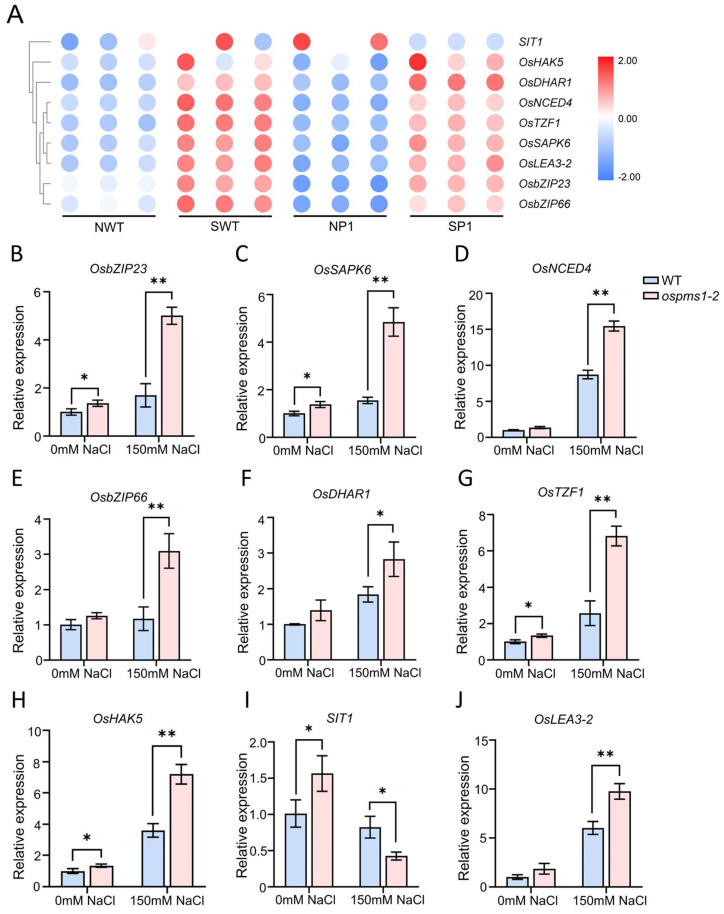
Differential expression and quantitative validation of salt stress response genes related to phytohormone signaling, reactive oxygen species, and osmotic balance in rice WT and *ospms1-2* mutant leaves under normal conditions and salt stress. (**A**) Clustering analysis of DEGs in WT and *ospms1*-2 mutants at 0 and 6 h of salt stress. Rows indicate the relative expression of individual genes, while columns indicate the level of expression in each sample. (**B**–**J**) Relative expression levels are expressed by fold change in wild-type expression levels relative to those before salt treatment (0 h). The housekeeping gene *OsActin* was used as an internal control. Data are shown as mean ± SD of at least three independent experiments. Significant differences were indicated by asterisks when compared with WT. (* *p* < 0.05 and ** *p* < 0.01, Students’ *t* test).

**Figure 9 genes-14-01621-f009:**
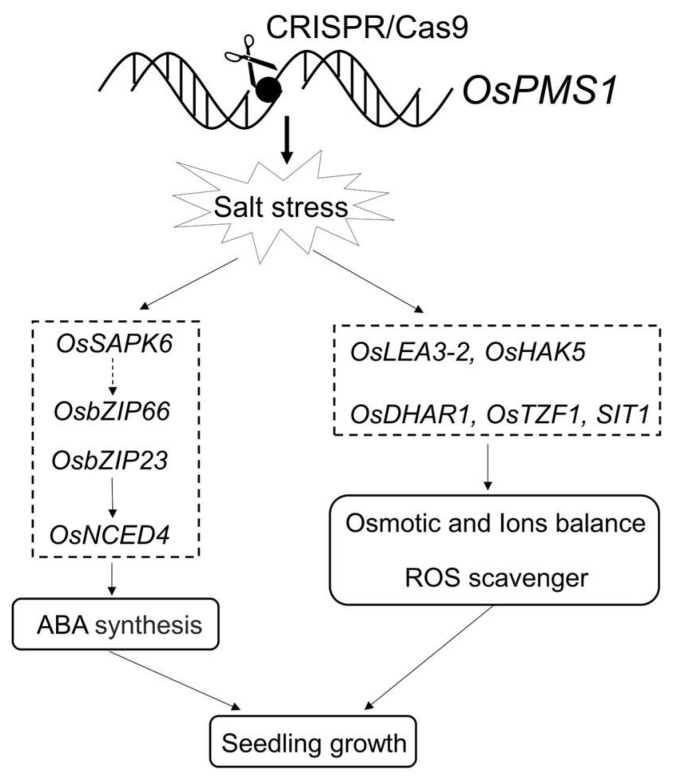
A possible mechanism model of *ospms1* mutant in response to salt stress at rice seedling stage. Solid arrows indicate confirmed pathways, based on literature reports and our work and dashed arrows indicate possible paths.

## Data Availability

The datasets generated for this study are available on request to the corresponding author.

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
