# Peer review of "OsPMS1 Mutation Enhances Salt Tolerance by Suppressing ROS Accumulation, Maintaining Na+/K+ Homeostasis, and Promoting ABA Biosynthesis"

_genes, 2023, doi:10.3390/genes14081621_

Round 1

Reviewer 1 Report

I really enjoyed it. This research is valuable. There I suggest to accept it directly as all aspects of work considered by authors. 

Author Response

Point-by-Point Response for Reviewers' comments:

Reviewer #1 (Remarks to the Author):

(1) I really enjoyed it. This research is valuable. There I suggest to accept it directly as all aspects of work considered by authors.

RESPONSE: Thank you for acknowledging the value of our work and we appreciate your positive comments.

Reviewer 2 Report

This is a well-written manuscript. In this study, the authors constructed the knockout lines of OsPMS1-ospms1, a critical gene in the rice DNA mismatch repair system. And then, the authors explored the molecular mechanism of ospms1 mutant under salt stress at physiological and transcriptome levels in comparison of OsPMS1. Finally, the study’s findings suggest that the ospms1 mutant tolerates salt stress at the seedling stage by inhibiting the accumulation of reactive oxygen species, maintaining Na+ and K+ homeostasis, and promoting ABA biosynthesis, which will provide a theoretical basis for breeding salt-tolerant rice varieties. 

Author Response

Point-by-Point Response for Reviewers' comments:

Reviewer #2 (Remarks to the Author):

(1) This is a well-written manuscript. In this study, the authors constructed the knockout lines of OsPMS1-ospms1, a critical gene in the rice DNA mismatch repair system. And then, the authors explored the molecular mechanism of ospms1 mutant under salt stress at physiological and transcriptome levels in comparison of OsPMS1. Finally, the study’s findings suggest that the ospms1 mutant tolerates salt stress at the seedling stage by inhibiting the accumulation of reactive oxygen species, maintaining Na+ and K+ homeostasis, and promoting ABA biosynthesis, which will provide a theoretical basis for breeding salt-tolerant rice varieties.

RESPONSE: We sincerely thank the reviewer for careful reading. We also thank you for acknowledging the value of our work and we appreciate your positive comments.

Reviewer 3 Report

The manuscript under review studies the role of OsPMS1, yet another gene involved in rice tolerance to abiotic stress. Despite the relevance of the research, current paper suffers from a number of issues that I will explain in detail below.

First of all, language in approximately 1/3 of the paper should be improved. Starting from the abstract, 33% of lines contain some inconsistent sentences.

For instance,

L16–L18: The ... study focused on the expression pattern ... and then generated ...

According to the Results section, the study focused on gene expression in OsPMS1 mutants and on physiological properties of these mutants.

L18–L19: ... and compared its phenotype ... and transcriptomic with ...

The word 'transcriptomic' is hanging without dependent words.

L23–L24: In contrast wild-type seedlings, ... seedlings of the MDA content ...

Nonsense phrases.

L27: ... ospms1 mutant treated seedlings with...

Strange word order.

L30: ... ascorbate and alternate metabolism ...

Please refer to what is discussed in section 3.6 of results – L390: aldarate metabolism.

L31: transcription-PCR osmoregulation (qRT-PCR)

What does that mean?

L34: ... ion transport (OsHAK5), and (OsLEA3-2).

... and what?

Simply put, these issues just in the Abstract indicate that at least some of the authors do not understand what they are writing about, and the others have not really read the manuscript. This impression is further strengthened with statement on L236 that 'OsPMS1 protein is 5,953 bp long ..., with 11 introns', or with attribution of AtPMS1 to Homo sapiens on L242–L243, or 'LEA' mentioned on L68 as a yet another class of organic molecules.

Materials and Methods section of an article is meant to provide the thorough details about the experimental section to facilitate the reproducibility of the research.

However, this section not only suffers from the abovementioned issues with language (check L100, L112, inconcistent verb tenses in L112, L115, L117, L118, L152, L168, L227–L228), but also contain sections of different level of detail. For instance, sections 2.1, 2.4.1,  2.4.2, 2.4.3, 2.4.4, 2.5.2 look as containing copied-and-pasted content from lab manuals 'as is'. Others (2.4.1, 2.4.3, 2.4.7) refer to manuals or kit instructions that are hard to obtain or even unavailable in English.

Section 2.2 refers to the paper [28] that indeed does not contain additional details, and only refers to (doi:10.1186/1746-4811-7-30) that finally contains some details on imaging but for another confocal microscope instrument. Direct information is expected on fluorochromes / fluorescent reporter proteins / reporter protein fusions, wavelengths applied for their excitation, and spectral windows used to detect their fluorescence.

In addition, mCherry protein mentioned on L131 looks just like a reporter protein, not a nuclear marker itself, doesn't it? The only hint is on L256 (AtWRKY25-mCherry) if correct indeed.

Section 2.4.3 refers to the paper [30] that itself does not contain any data on chlorophyll measurement with SPAD meter. Meanwhil, section 2.4.3 postulates that SPAD meter is a simple and portable instrument, which does not really add a sense. Neither section 2.4.3 nor the caption for Figure 2E clarify what are the units and/or normalization factor for chlorophyll content.

Finally, section 2.6 of Methods declares that 'All data in this paper ... are represented by error bars...' (language!), that error bars show either SD or standard error however figures employ only SD on histograms. Authors are welcome to explain why one-way ANOVA is appropriate test in case of small sample sizes.

Results section contains less language issues (nevertheless pay attention to L243, L246, L247–L248, L318, L384–L385), however needs improvement. For instance, Fig. 1C does not indicate any reason why mCherry served as a nuclear marker (see above). Fig. 1B referred from L251 does not show any marks for significant increase in OsPMS1 relative expression.

Fig. 2E should indicate which units are employed to express chlorophyll content. Section 2.4.3 of Materials and Methods currently does not serve for that.

Authors should explain what do they mean under 'the biosynthetic pathway of metabolic pathways' (L384–L385). Also on L387 'MAPK signalling pathway-plant' looks wierd since the article itself is focused on plants. Finally it is important to substantiate the choice of 9 candidate genes among all DEGs for further screening (L397–L398). Here, moving the Fig. S2 to the main text and highlighting these genes in volcano plot would be a step towards explanatory figure instead of a standard report image.

Table S2 does not provide information what type of data are represented, mean±SD or mean±std.error.

Discussion section is limited to the number of studied genes and physiological features and does not attempt to move forward and discuss at least why the protein employed in DNA reparation appears to be involved in salt stress response. Further, Fig. 8 is more likely to show the time-course of relationship between actors of salt stress response but not the 'molecular mechanism'. Well, we will follow authors' narrow scope and touch only what is discussed. Authors should pay attention to that the statement 'the soluble protein content of the ospms1 mutant was higher than that of the wild-type plant' on L441 was not supported by data (Fig. 3A). 

Authors consider ROS burst and osmotic effects of salt stress in plant as different and distant processes (Fig. 8). However, these processes seem to be tightly interrelated, and ion fluxes to be partly regulated by ROS. See, for instance, Demidchik et al., 2014 (doi:10.1093/jxb/eru004). For instance, in Arabidopsis root growth arrest under 100mM NaCl salt stress, osmotic effects are responsible for merely 20% of the effect while the rest is governed by ROS.

In summary, authors are expected to provide the bigger picture of the salt stress tolerance mechanisms in rice and the putative role of OsPMS1. That would indicate whether the paper is timely and give any novelty.

English Language should be addressed very attentively, especially for the Abstract, Introduction and Materials and Methods sections. Please refer to comments above.

Author Response

Point-by-Point Response for Reviewers' comments:

Reviewer #3 (Remarks to the Author):

The manuscript under review studies the role of OsPMS1, yet another gene involved in rice tolerance to abiotic stress. Despite the relevance of the research, current paper suffers from a number of issues that I will explain in detail below.

RESPONSE: We feel great thanks for your professional review work on our article. According to your nice suggestions, we have made extensive corrections to our previous manuscript. We addressed your comments in detail point-by-point as follows.

(1) First of all, language in approximately 1/3 of the paper should be improved. Starting from the abstract, 33% of lines contain some inconsistent sentences.

RESPONSE: We like to thank you for your comments about our manuscript, which gave us guidelines and thus significantly improve the quality of our manuscript.

(2) L16–L18: The ... study focused on the expression pattern ... and then generated ...

According to the Results section, the study focused on gene expression in OsPMS1 mutants and on physiological properties of these mutants.

RESPONSE: Thank you for pointing out the inappropriateness of this sentence. We have changed to “The present study focused on the expression pattern of the rice mismatch repair gene post-meiotic segregation 1 (OsPMS1), studied the physiological properties and performed transcriptome analysis of ospms1 mutant seedlings in response to salt stress”.

(3) L18–L19: ... and compared its phenotype ... and transcriptomic with ...

The word 'transcriptomic' is hanging without dependent words.

RESPONSE: Thank you for pointing this out, which has been amended in the revised manuscript.

(4) L23–L24: In contrast wild-type seedlings, ... seedlings of the MDA content ...

Nonsense phrases.

RESPONSE: Thank you for your careful review. This word is really redundant, we have deleted it in the revised manuscript.

(5) L27: ... ospms1 mutant treated seedlings with...

Strange word order.

RESPONSE: We were really sorry for our careless mistakes. Thank you for your reminder. We have replaced the correct word order as "ospms1 mutant seedlings treated with…".

(6) L30: ... ascorbate and alternate metabolism ...

Please refer to what is discussed in section 3.6 of results – L390: aldarate metabolism.

RESPONSE: Thank you for pointing out this mistake. We are sorry for our carelessness. Based on your comments we have made changes to harmonize the wording of ascorbate and aldarate metabolism throughout the manuscript.

(6) L31: transcription-PCR osmoregulation (qRT-PCR)

What does that mean?

RESPONSE: We sincerely thank the reviewer for careful reading. As suggested by the reviewer, we have corrected the "quantitative real-time reverse transcription-PCR osmoregulation (qRT-PCR)" into "quantitative real-time reverse transcription-PCR (qRT-PCR)."

(7) L34: ... ion transport (OsHAK5), and (OsLEA3-2).

... and what?

RESPONSE: Thanks for your careful review. We have added the term "osmoregulation" before the OsLEA3-2 gene in the revised manuscript.

(8) Simply put, these issues just in the Abstract indicate that at least some of the authors do not understand what they are writing about, and the others have not really read the manuscript. This impression is further strengthened with statement on L236 that 'OsPMS1 protein is 5,953 bp long ..., with 11 introns', or with attribution of AtPMS1 to Homo sapiens on L242–L243, or 'LEA' mentioned on L68 as a yet another class of organic molecules.

RESPONSE: We were really sorry for our careless mistakes. Thank you for your reminder. We have corrected these errors in the revised manuscript.

(9) Materials and Methods section of an article is meant to provide the thorough details about the experimental section to facilitate the reproducibility of the research.

RESPONSE: Your suggestion is very helpful to me and I have added some details of the experiment in the materials and methods.

(10) However, this section not only suffers from the abovementioned issues with language (check L100, L112, inconcistent verb tenses in L112, L115, L117, L118, L152, L168, L227–L228), but also contain sections of different level of detail. For instance, sections 2.1, 2.4.1,  2.4.2, 2.4.3, 2.4.4, 2.5.2 look as containing copied-and-pasted content from lab manuals 'as is'. Others (2.4.1, 2.4.3, 2.4.7) refer to manuals or kit instructions that are hard to obtain or even unavailable in English.

RESPONSE: Thank you for pointing out our language and grammatical errors, which we have corrected in the revised manuscript.

(11) Section 2.2 refers to the paper [28] that indeed does not contain additional details, and only refers to (doi:10.1186/1746-4811-7-30) that finally contains some details on imaging but for another confocal microscope instrument. Direct information is expected on fluorochromes / fluorescent reporter proteins / reporter protein fusions, wavelengths applied for their excitation, and spectral windows used to detect their fluorescence.

RESPONSE: We apologize for the error in the reference. We have added direct information on the wavelength of the excitation and the wavelength of the detection light in the revised manuscript.

(12) In addition, mCherry protein mentioned on L131 looks just like a reporter protein, not a nuclear marker itself, doesn't it? The only hint is on L256 (AtWRKY25-mCherry) if correct indeed.

RESPONSE: Thank you for your careful review. Indeed, mCherry is a monomeric red fluorescent protein and which we have corrected.

(13) Section 2.4.3 refers to the paper [30] that itself does not contain any data on chlorophyll measurement with SPAD meter. Meanwhil, section 2.4.3 postulates that SPAD meter is a simple and portable instrument, which does not really add a sense. Neither section 2.4.3 nor the caption for Figure 2E clarify what are the units and/or normalization factor for chlorophyll content.

RESPONSE: Thank you for your careful review. This sentence is really redundant, we have changed the reference and deleted it in the revised manuscript. When the SPAD-502 Chlorophyll Meter is measuring the absorbance of a leaf in the red region and in the near infrared region, it can be used to calculate a kind of SPAD value, which is a numerical representation of a parameter that currently corresponds to the chlorophyll content in the leaf. The unit of relative chlorophyll content is SPAD.

(14) Finally, section 2.6 of Methods declares that 'All data in this paper ... are represented by error bars...' (language!), that error bars show either SD or standard error however figures employ only SD on histograms. Authors are welcome to explain why one-way ANOVA is appropriate test in case of small sample sizes.

RESPONSE: We are very lucky that you pointed out this mistake. Indeed, it should be changed to “All data in this paper were expressed as mean ± standard deviation (SD). The error bars indicate the SD.” We are also grateful to you for pointing out our conceptual error that requires a larger sample size (n > 30) when using one-way ANOVA, while Student's t test is mainly used for smaller sample sizes (n < 30). We used the same two sets of data to obtain the same P-value by Student's t test and one-way ANOVA, so we changed it to Student’s t test in the revised manuscript.

(15) Results section contains less language issues (nevertheless pay attention to L243, L246, L247–L248, L318, L384–L385), however needs improvement. For instance, Fig. 1C does not indicate any reason why mCherry served as a nuclear marker (see above). Fig. 1B referred from L251 does not show any marks for significant increase in OsPMS1 relative expression.

RESPONSE: Thank you for pointing out these language issues. We have corrected the description and added more details to the legend of Figure 1C. Our description of Figure 1B was incorrect, and we have changed it to “The qRT-PCR analysis showed that the expression of OsPMS1 can be induced by salt stress”.

(16) Fig. 2E should indicate which units are employed to express chlorophyll content. Section 2.4.3 of Materials and Methods currently does not serve for that.

RESPONSE: Thank you for your advice. We have rewritten Section 2.5.3 of Materials and Methods in the revised manuscript.

(17) Authors should explain what do they mean under 'the biosynthetic pathway of metabolic pathways' (L384–L385). Also on L387 'MAPK signalling pathway-plant' looks wierd since the article itself is focused on plants. Finally it is important to substantiate the choice of 9 candidate genes among all DEGs for further screening (L397–L398). Here, moving the Fig. S2 to the main text and highlighting these genes in volcano plot would be a step towards explanatory figure instead of a standard report image.

RESPONSE: Sorry, our statement is not accurate. We have changed to "The DEGs obtained from the SWT vs. NWT group and the SP1 vs. NP1 group significantly enriched metabolic pathways and biosynthesis of secondary metabolites (Figure 7A, B)." Thank you for this valuable suggestion. We agree that 'MAPK signaling pathway-plant' looks weird. Thus, we removed it and moved Fig. S2 to the main text, highlighting these genes in the volcano plot in the revised manuscript.

(18) Table S2 does not provide information what type of data are represented, mean ±SD or mean ± std.error.

RESPONSE: Thank you for your advice. We have noted for Table S2 that results are the mean ± SD of three replicates. (*p < 0.05 and **p < 0.01, Students' t test).

(19) Discussion section is limited to the number of studied genes and physiological features and does not attempt to move forward and discuss at least why the protein employed in DNA reparation appears to be involved in salt stress response. Further, Fig. 8 is more likely to show the time-course of relationship between actors of salt stress response but not the 'molecular mechanism'. Well, we will follow authors' narrow scope and touch only what is discussed. Authors should pay attention to that the statement 'the soluble protein content of the ospms1 mutant was higher than that of the wild-type plant' on L441 was not supported by data (Fig. 3A).

RESPONSE: Thank you for your comment. We agree that the statement 'the soluble protein content of the ospms1 mutant was higher than that of the wild-type plant' on L441 was not supported by data. We thus removed it in the revised manuscript.

(20) Authors consider ROS burst and osmotic effects of salt stress in plant as different and distant processes (Fig. 8). However, these processes seem to be tightly interrelated, and ion fluxes to be partly regulated by ROS. See, for instance, Demidchik et al., 2014 (doi:10.1093/jxb/eru004). For instance, in Arabidopsis root growth arrest under 100mM NaCl salt stress, osmotic effects are responsible for merely 20% of the effect while the rest is governed by ROS.

RESPONSE: Thank you for your proposal. Indeed, when plants are subjected to salt stress, osmotic regulation, ion balance, and ROS scavenging are closely related, so we have made corresponding modifications in Figure 9 of the revised manuscript.

(21) In summary, authors are expected to provide the bigger picture of the salt stress tolerance mechanisms in rice and the putative role of OsPMS1. That would indicate whether the paper is timely and give any novelty.

RESPONSE: Thank you for your excellent advice. At present, there are limited reports on the role of rice MMR protein in response to salt stress. We initially investigated the mechanism of the rice mismatch repair protein OsPMS1 in response to salt stress at the seedling stage. We consider this to be novel. Although the transcriptome data screened some of the differentially expressed genes, the specific functions of these genes still need to be verified. Therefore, in future studies, we will work towards validating functional genes and identifying salt tolerance during the whole growth period to provide a bigger picture of the mechanisms involved in salt stress of OsPMS1.

(22) English Language should be addressed very attentively, especially for the Abstract, Introduction and Materials and Methods sections. Please refer to comments above.

RESPONSE: Thank you again for your careful review and valuable suggestions to improve the quality of our manuscript. We have revised the manuscript according to your comments.

Reviewer 4 Report

Dear Authors,

You will find all comments and reccomendations on the manuscript itself.

Best Regards

Author Response

Point-by-Point Response for Reviewers' comments:

Reviewer #4 (Remarks to the Author):

(1) L98-101: Describe a bit more the aim of the research.

RESPONSE: Your suggestion is very helpful to me and we have redescribed the aim of the research a bit in the revised manuscript.

(2) L129-130: Use Italic style for writting species names.

RESPONSE: Thanks for your careful review. We are sorry for our carelessness. Based on your comments we have made changes to the writing of typically species names, genes, and ions.

(3) L143: solution(pH 7.2) Add a space.

RESPONSE: Thank you for your careful review. We have added a a space.

(4) L170:Enter this formula inside of the text.

RESPONSE: Thank you for pointing this out, which has been amended in the revised manuscript.

(5) L211-213: it is better to “The SOAP nuke software was used to remove reads with un-known bases (> 5%) and low-quality reads (…) in order to obtain a clean read [35]”

RESPONSE: Thank you for this thoughtful comment, which we have corrected.

(6) L211: Oryza sativa should Use Italic style.

RESPONSE: We are very lucky that you pointed out this mistake, which we have corrected.

(7) Please check the manuscript the Latin mane of the plant species.

RESPONSE: Thank you for pointing out this mistake, which we have corrected.

(8) L492: There is no need to put the figure here as you already discussed them in the results section.

RESPONSE: Thank you for your advice. We have deleted it in the revised manuscript.

(9) Re-check carefully the reference section and follow journal's rules to write it.

RESPONSE: Thank you again for your careful review and valuable suggestions. We have revised the format of the references according to the rules of the journal.

Reviewer 5 Report

There are several typographical errors that need to be addressed relating to language and the appropriate use of scientific convention. This varies significantly across the manuscript, with some sections being clearer than others. The materials and methods specifically need improvement.

Typically species names and genes are presented in italics which improves clarity to the reader. Ions typically are noted in superscript in scientific manuscripts.

I am not familiar with all of the techniques that the authors employed, the manuscript is missing context that explains why different techniques were employed and how this contributes to their findings. Some of this information is presented in the results when it is better suited for the methods to improve clarity and logical flow. Considerable digging was needed using external sources to interpret their workflow which should be readily apparent in their methods. Most readers will not search this information out.

There is inconsistency in referencing OsPMS1, in some sections and figures it is unclear if they are referring to the mutant or wild-type (for example on line 106 it reads as they are generating mutants of ospms1?).

Please check names of equipment and reagents such as devices such as the Nanodrop 2000 and Agilent 2100 are not spelt “Nanodrop 2,000” or “Agilent 2,100”

Please check for common spelling errors throughout such as “indicia” vs. “indica”

The quality of the materials and methods needs improvement, it is not clear why the authors selected their given techniques. Further context is needed to connect the introduction to the experimental design.

It is not clear how the techniques tie into the overall experiment, the various subsections of the methods could be improved with brief statements such as “to determine X, an indicator of Y, we employed method Z”. This occurs no fewer than 4 times in the manuscript and reduces readability.

The authors do not describe several of their methods and rely heavily on “as previously described”, this leaves out context and reduces reproducibility. It would not be possible to reproduce the experiment following the description the authors have provided. Taking shortcuts makes the work less accessible to the reader.

Section 3.1 does not read as a results section but a mixture of introduction, methods and discussion. The authors need to be careful to not get side-tracked in presenting the results.

The quality of the writing is highly variable across the manuscript. Several sections include odd or unclear wording. 

There are many typographical errors relating to basic scientific nomenclature that reduce readability.

Sections of the results appear to be better suited for the introduction, discussion, or conclusions and not the results. 

Overall clarity is low in the methods section but other sections appear fine. 

Author Response

Point-by-Point Response for Reviewers' comments:

Reviewer #5 (Remarks to the Author):

(1) There are several typographical errors that need to be addressed relating to language and the appropriate use of scientific convention. This varies significantly across the manuscript, with some sections being clearer than others. The materials and methods specifically need improvement.

RESPONSE: We feel great thanks for your professional review work on our article. According to your nice suggestions, we have made extensive revisions to the materials and methods section of the previous manuscript.

(2) Typically species names and genes are presented in italics which improves clarity to the reader. Ions typically are noted in superscript in scientific manuscripts.

RESPONSE: Thanks for your careful review. We are sorry for our carelessness. Based on your comments we have made changes to the writing of typically species names, genes, and ions.

(3) I am not familiar with all of the techniques that the authors employed, the manuscript is missing context that explains why different techniques were employed and how this contributes to their findings. Some of this information is presented in the results when it is better suited for the methods to improve clarity and logical flow. Considerable digging was needed using external sources to interpret their workflow which should be readily apparent in their methods. Most readers will not search this information out.

RESPONSE: Thanks for your valuable suggestions, we have moved the relevant information from the results section to the materials and methods section in the revised manuscript.

(4) There is inconsistency in referencing OsPMS1, in some sections and figures it is unclear if they are referring to the mutant or wild-type (for example on line 106 it reads as they are generating mutants of ospms1?).

RESPONSE: Thank you for your careful review, which we have corrected.

(5) Please check names of equipment and reagents such as devices such as the Nanodrop 2000 and Agilent 2100 are not spelt “Nanodrop 2,000” or “Agilent 2,100”

RESPONSE: Thank you for pointing out this mistake. We have corrected the "NanoDrop 2,000" into "NanoDrop 2000" and the "Agilent Bioanalyzer 2,100" into "Agilent Bioanalyzer 2100".

(6) Please check for common spelling errors throughout such as “indicia” vs. “indica”

RESPONSE: We were really sorry for our common spelling errors. We have changed as “indica”.

(7) The quality of the materials and methods needs improvement, it is not clear why the authors selected their given techniques. Further context is needed to connect the introduction to the experimental design.

RESPONSE: Thank you for your valuable suggestions, in the Materials and Methods section of the revised manuscript we have explained in a short statement why some of the experiments were carried out.

(8) It is not clear how the techniques tie into the overall experiment, the various subsections of the methods could be improved with brief statements such as “to determine X, an indicator of Y, we employed method Z”. This occurs no fewer than 4 times in the manuscript and reduces readability.

RESPONSE: Thank you for your nice advice. Indeed, from a contextual point of view, the reading after the revision is clearer, and we have modified it.

(9) The authors do not describe several of their methods and rely heavily on “as previously described”, this leaves out context and reduces reproducibility. It would not be possible to reproduce the experiment following the description the authors have provided. Taking shortcuts makes the work less accessible to the reader.

RESPONSE: Thank you for pointing out our errors, we have described some of the experimental techniques in detail in the revised manuscript.

(10) Section 3.1 does not read as a results section but a mixture of introduction, methods and discussion. The authors need to be careful to not get side-tracked in presenting the results.

RESPONSE: We sincerely thank the reviewer for careful reading. In response to your suggestions, we have made changes to section 3.1.

(11) The quality of the writing is highly variable across the manuscript. Several sections include odd or unclear wording.

RESPONSE: Thank you for your advice. We have changed the description in the revised manuscript.

(12) There are many typographical errors relating to basic scientific nomenclature that reduce readability.

RESPONSE: We were really sorry for our careless mistakes. Thank you for your reminder. We have corrected in the revised manuscript.

(13) Sections of the results appear to be better suited for the introduction, discussion, or conclusions and not the results.

RESPONSE: Thank you for pointing this out, which has been amended in the revised manuscript.

(14) Overall clarity is low in the methods section but other sections appear fine. 

RESPONSE: Thank you for your positive comments, we have focused on the materials and methods section in the revised manuscript.

Round 2

Reviewer 3 Report

This is my second review of the manuscript. Authors took an effort to improve the manuscript, especially in Materials and Methods, Abstract and in a few figures. However, the newly added parts contain some flaws that I suggest to fix or improve. I put these remarks below without specific sorting into language-specific and general remarks and ask the authors to take this graciously. Hopefully dealing with these would be the last round of the review process.

L24: Incorrect superscript next to the symbols for superoxide anion radical. Please take an effort to refer to e.g. Wikipedia.

L109–L114: these rewritten lines still need some language improvement (prepositions, correct tenses)

L115–L116: "containing 150 mM NaCl under pressure for 10 days"

What does it mean "under pressure"?

L119–L120: incorrect tense

L149: Incorrect symbol for superoxide anion radical (see above)

L150, L155: incorrect tense

L267: OsPMS1 genome ?!

Fig. 1C,D: please follow your corrections in the manuscript and figure caption: label the second column of fluorescence images as "AtWRKY25-mCherry", not just "mCherry".

Fig. 9 (L558): correct the typo in the last block (should read as "Seedling growth").

L109–L114: these rewritten lines still need some language improvement (prepositions, correct tenses)

L119–L120: incorrect tense

L150, L155: incorrect tense

Fig. 9 (L558): correct the typo in the last block (should read as "Seedling growth").

Author Response

Point-by-Point Response for Reviewers' comments:

Reviewer #3 (Remarks to the Author):

This is my second review of the manuscript. Authors took an effort to improve the manuscript, especially in Materials and Methods, Abstract and in a few figures. However, the newly added parts contain some flaws that I suggest to fix or improve. I put these remarks below without specific sorting into language-specific and general remarks and ask the authors to take this graciously. Hopefully dealing with these would be the last round of the review process.

RESPONSE: Thank you again for your careful review and valuable suggestions. We have revised the manuscript according to your comments.

(1) L24: Incorrect superscript next to the symbols for superoxide anion radical. Please take an effort to refer to e.g. Wikipedia.

RESPONSE: Thank you for pointing out this mistake, which we have corrected.

(2) L109–L114: these rewritten lines still need some language improvement (prepositions, correct tenses)

RESPONSE: We are very lucky that you pointed out this mistake, which we have corrected.

(3) L115–L116: "containing 150 mM NaCl under pressure for 10 days" What does it mean "under pressure"?

RESPONSE: Thank you for your careful review. We have deleted it in the revised manuscript.

(4) L119–L120: incorrect tense

RESPONSE: Thank you for pointing this out, which has been amended in the revised manuscript.

(5) L149: Incorrect symbol for superoxide anion radical (see above)

RESPONSE: Thank you for pointing out this mistake, which we have corrected.

(6) L150, L155: incorrect tense

RESPONSE: Thank you for pointing out our tense errors, which we have corrected in the revised manuscript.

(7) L267: OsPMS1 genome ?!

RESPONSE: Sorry, our statement is not accurate, which we have corrected in the revised manuscript.

(8) Fig. 1C,D: please follow your corrections in the manuscript and figure caption: label the second column of fluorescence images as "AtWRKY25-mCherry", not just "mCherry".

RESPONSE: Thank you for your excellent advice. We have made changes to the figure 1 in the revised manuscript.

(9) Fig. 9 (L558): correct the typo in the last block (should read as "Seedling growth").

RESPONSE: We are very lucky that you pointed out this mistake, which we have corrected.

Reviewer 5 Report

The authors have done a good job addressing the reviewer's comments, further work is needed to improve grammar for clarity but otherwise an improvement. 

Tense needs to be reviewed to improve clarity.

Author Response

Point-by-Point Response for Reviewers' comments:

Reviewer #5 (Remarks to the Author):

The authors have done a good job addressing the reviewer's comments, further work is needed to improve grammar for clarity but otherwise an improvement. 

RESPONSE: Thank you for recognition us and we have made changes to the tenses based on your suggestions.

(1) Tense needs to be reviewed to improve clarity.

RESPONSE: We are very lucky that you pointed out this mistake, which we have corrected.